# Understanding Acanthamoeba Keratitis: An In-Depth Review of a Sight-Threatening Eye Infection

**DOI:** 10.3390/microorganisms12040758

**Published:** 2024-04-09

**Authors:** Francesco Petrillo, Antonia Tortori, Veronica Vallino, Marilena Galdiero, Antonio M. Fea, Ugo De Sanctis, Michele Reibaldi

**Affiliations:** 1Department of Surgical Sciences, University of Turin, 10126 Turin, Italy; veronica.vallino@gmail.com (V.V.); antoniofea@icloud.com (A.M.F.); ugo.desanctis@unito.it (U.D.S.); mreibaldi@libero.it (M.R.); 2Department of Ophthalmology, “City of Health and Science” Hospital, 10126 Turin, Italy; 3Ophthalmology Unit, Surgery Department, Piacenza Hospital, 29121 Piacenza, Italy; antonia.tortori@gmail.com; 4Department of Experimental Medicine, University of Campania “Luigi Vanvitelli”, 81100 Naples, Italy; marilena.galdiero@gmail.com

**Keywords:** acanthamoeba, keratitis, contact lens, acanthamoeba keratitis, corneal ulcer

## Abstract

Acanthamoeba keratitis (AK) is a rare but potentially sight-threatening corneal infection caused by the Acanthamoeba parasite. This microorganism is found ubiquitously in the environment, often in freshwater, soil, and other sources of moisture. Despite its low incidence, AK presents significant challenges due to delayed diagnosis and the complex nature of therapeutic management. Early recognition is crucial to prevent severe ocular complications, including corneal ulceration and vision loss. Diagnostic modalities and treatment strategies may vary greatly depending on the clinical manifestation and the available tools. With the growing reported cases of Acanthamoeba keratitis, it is essential for the ophthalmic community to thoroughly understand this condition for its effective management and improved outcomes. This review provides a comprehensive overview of AK, encompassing its epidemiology, risk factors, pathophysiology, clinical manifestations, diagnosis, and treatment.

## 1. Introduction

Microbial keratitis, also referred to as infectious corneal ulcers, arises from the proliferation of various microorganisms including bacteria, fungi, viruses, and parasites. This proliferation induces the inflammation and degradation of the corneal tissue, thereby endangering visual acuity and presenting a significant challenge in the field of ophthalmology [1,2,3,4,5]. Among these infections, Acanthamoeba keratitis stands out as a parasitic corneal affliction provoked by a free-living amoeba commonly inhabiting water and soil environments. While this condition is infrequent, its potential to affect vision is noteworthy, with documented cases escalating annually on a global scale [5]. The objective of our manuscript is to comprehensively explore all aspects of Acanthamoeba keratitis, encompassing its epidemiological characteristics, risk factors, pathophysiology, clinical manifestations, diagnostic investigations, and therapeutic management. Our aim is to provide an exhaustive overview of this condition with the ultimate goal of advancing understanding and management strategies for this burgeoning ophthalmological concern.

## 2. Incidence

Since its initial documentation in 1973 [6], Acanthamoeba keratitis (AK) has been recognized as a relatively rare infection characterized by sporadic outbreaks [5]. The frequency of AK has exhibited fluctuations, often demonstrating an upward trend, with outbreaks typically confined to specific geographical regions. The widespread adoption of contact lenses worldwide during the 1980s has been associated with the escalating global incidence of AK [7]. Quantifying the incidence of AK poses challenges, which can be approached by considering the total number of contact-lens wearers, the demographic most vulnerable to infection, or by assessing it relative to the entire population. However, both methodologies are subject to limitations, as many estimations rely on analyses of specific outbreak scenarios [5]. The global annual prevalence of Acanthamoeba keratitis (AK) stands at 23,561 cases, corresponding to a rate of 2.9 cases per million individuals. Notably, the highest documented incidence is recorded in India, with 15.2 cases per million people, while Tunisia and Belgium exhibit the lowest at 0.2 cases per million individuals [7]. In the United States, the incidence of AK witnessed a steady increase from 1999 to 2007, rising from 0.4 to 0.8 cases per million people [8]. Within the Chicago area between 2003 and 2005, AK outbreaks were documented, estimating an incidence of 1.3 cases per million individuals, with notable disparities in distribution, potentially influenced by a reduced availability of disinfection products due to Environmental Protection Agency (EPA) regulations [9]. Furthermore, there was a notable surge in AK incidence in Philadelphia and Houston in 2004, peaking at approximately 1.9 and 2.2 cases per million people, respectively [7].

In British Columbia, Canada, the incidence of Acanthamoeba keratitis (AK) surged from 0.3 to 2.0 cases per million individuals between 1995 and 2005, aligning with an AK outbreak in Toronto in 2003, where the incidence also reached 2.0 cases per million people [10]. This incidence trend in Canada paralleled that of the United States, both experiencing a notable upsurge in AK incidence from 2003 compared to preceding years. Within Europe, the United Kingdom exhibited the highest AK incidence, likely attributed to the widespread use of contact lenses since 1992 [11]. In Bristol, AK incidence appeared to escalate post-1990, peaking at approximately 7.92 cases per million people. Encouragingly, there was a decline in AK cases between 1995 and 1996 in England, with incidence dropping to 1.4 cases per million individuals. Conversely, West Scotland reported an exceptionally elevated AK incidence of around 7.0 cases per million people. In Wales, AK incidence fluctuated between 1.3 and 1.1 cases per million individuals from 1997 to 1999. Moreover, Coventry in England experienced a substantial increase in AK incidence from 2017 to 2018 [12,13,14]. Randag et al. have conducted a comprehensive assessment of Acanthamoeba keratitis (AK) incidence within the Netherlands spanning from 2009 to 2015. Their study revealed a notable escalation in AK cases, rising from 16 instances in 2009 to 45 occurrences in 2015. This increase signifies a substantial rise in AK incidence, translating to approximately one case per 21,000 contact lens wearers in 2015 [15]. Nielsen et al. conducted a retrospective study at the tertiary ophthalmology department of Aarhus University Hospital in Denmark. From 1994 to 2018, they observed a notable increase in the incidence of Acanthamoeba keratitis (AK), with rates escalating from 0.13 cases per million per year during the initial five years to 2.7 cases per million per year in the last five years of the study period [16]. In Oceania, the incidence of AK averaged approximately 0.5–3.1 cases per million individuals during infection outbreaks spanning from 1990 to 2016. Across Asia, Coimbatore, India, reported an exceptionally high incidence of AK at 17.6 cases per million people from 2017 to 2019, with some patients experiencing co-infections with fungi and bacteria. Madurai, India, witnessed an increase in AK incidence in 1993 and 2002, estimated at 13.1 cases per million people. China, Singapore, and Israel reported comparable incidence rates at 0.6, 1.1, and 0.6 cases per million people, respectively. Acanthamoeba keratitis (AK) remains comparatively less prevalent in comparison to other forms of microbial keratitis associated with contact-lens use or unrelated to it. Contact-lens-related microbial keratitis typically occurs at rates ranging from 1 to 25 cases per 10,000 wearers per year, a figure largely influenced by wearing regimens. In contrast, AK presents with an incidence as high as 1 in 30,000 contact-lens users annually. Notably, in India, Acanthamoeba was detected in 1% of all patients cultured for potential infectious keratitis, irrespective of the underlying cause [5,17,18,19,20,21]. The occurrence of AK affecting both eyes in contact-lens wearers is rare, observed in only 4% to 11% of AK cases. This limited occurrence of bilateral AK, alongside the significant population susceptible to AK due to contact-lens usage, suggests the involvement of factors beyond mere contact-lens wear in the development of AK [22]. Investigating these additional risk factors and gaining insights into the pathogenic mechanisms underlying AK are crucial endeavors in understanding and addressing this condition.

## 3. Risk Factors

The risk factors associated with Acanthamoeba keratitis (AK) exhibit variability across different countries [7]. Among otherwise healthy individuals of any age, the predominant risk factor for AK is the use of contact lenses, with more than 90% of cases linked to such use [23]. While the risk of bacterial keratitis may be lower in the users of rigid lenses, this does not hold true for amoebic keratitis. The resurgence of rigid lenses for orthokeratology has accentuated an increased risk, particularly in cases involving factors such as epithelial thinning, inadequate hygiene practices, and exposure to tap water during lens-care routines [24]. Both soft- and rigid-lens wearers encounter heightened risks associated with factors like ocular trauma, swimming while wearing lenses, and noncompliance with recommended disinfection protocols. Additional risks include wearing lenses during activities such as hot-tub use and rinsing lenses or their cases with non-sterile water. Another significant determinant influencing the onset of AK is the use of disinfectant solutions that lack efficacy against Acanthamoeba [5,6,7]. It is noteworthy that Acanthamoeba cysts display high resistance to chlorine and many of the current multipurpose solutions (MPSs) available. Actually, in the United Kingdom, the surge in AK cases followed the introduction of chlorine-based disinfectant systems. The subsequent decrease in incidence observed from 1995 to 1996 is likely attributable to advancements in lens-care practices and the implementation of more effective disinfection systems [25]. Moreover, the escalation of Acanthamoeba keratitis in the United States may also be attributed to the inadequate efficacy of certain solutions against Acanthamoeba, notably AMOCMP [26]. The most effective systems for combating both forms of Acanthamoeba involve heat and hydrogen peroxide disinfection, especially a two-step process ensuring prolonged exposure before neutralization. In developing nations, trauma remains the primary risk factor for Acanthamoeba keratitis (AK), accounting for an estimated 27% of cases. A retrospective analysis conducted in India from 1999 to 2002 revealed that all 33 AK patients were agricultural workers who did not use contact lenses, with each having a history of ocular trauma [27]. Similarly, a study conducted in China from 1997 to 2003 indicated that 50.8% of patients (96 out of 189) were farmers. Additional significant risk factors include water contamination and warmer climates [28]. In Australia, the reduced incidence of Acanthamoeba keratitis (AK) in Melbourne is likely attributed to cleaner water supplies. Furthermore, an observed surge in Acanthamoeba keratitis (AK) cases during the period between June and August in England, Hong Kong, and Canada suggests a seasonal pattern. This phenomenon could be attributed to amoebas thriving in warmer environments, with heightened exposure during activities such as swimming and bathing in the summer [29]. Geographical disparities in AK incidence in the UK may also be influenced by water hardness; individuals supplied with hard water (containing 200 milligrams or more of calcium carbonate per liter) exhibited a threefold higher risk of AK compared to those with soft water (containing 0–99 milligrams of calcium carbonate per liter) [30]. Additional minor risk factors for AK encompass steroid use, topical anesthetic application, co-infections with other microorganisms, complications associated with systemic diseases, and the use of cosmetics [5].

Understanding the primary risk factors associated with Acanthamoeba keratitis (AK) is pivotal for infection prevention. Mitigating risky behaviors, enhancing water-sanitation practices, and promptly seeking care at specialized corneal disease centers can significantly contribute to preventing AK infection or its most severe complications [15].

## 4. Classification

Acanthamoebae are ubiquitous organisms found freely inhabiting water and soil, where they prey upon other microorganisms. They exhibit two distinct forms: a vulnerable and motile trophozoite, and a resilient double-walled cyst capable of withstanding harsh environmental conditions such as temperature fluctuations, desiccation, irradiation, antimicrobial agents, and shifts in environmental parameters. When confronted with adverse circumstances, trophozoites swiftly encyst, thereby preserving their capacity to generate viable trophozoites for extended periods, sometimes spanning several years [1,31,32,33,34].

To organize the expanding array of isolates belonging to the Acanthamoeba genus, Pussard and Pons (1977) undertook an initial classification of the species through the examination of the morphological characteristics of the cysts. Following this, the ectocyst and endocyst shapes and sizes were taken into account, leading to the segregation of Acanthamoeba spp. into 24 species [35,36,37]. Among the various Acanthamoeba species implicated in keratitis, *Acanthamoeba castellanii* stands out as the most commonly associated, with *Acanthamoeba polyphaga* and *Acanthamoeba hatchetti* also frequently identified. Other species, including *Acanthamoeba culbertsoni*, *Acanthamoeba rhysodes*, *Acanthamoeba lugdunesis*, *Acanthamoeba quina*, and *Acanthamoeba griffini*, have also been described in relation to keratitis cases [1,32]. However, the current reliance on morphological criteria for classification is deemed to be ambiguous and unreliable. Variations in species morphology can occur due to differences in culture media conditions, resulting in inconsistencies in cyst morphology, which is pivotal for species identification. Furthermore, a multitude of studies have identified disparities in the cyst morphology of genetically identical isolates, underscoring the inadequacy of relying solely on morphological characteristics for species identification and highlighting the necessity for molecular methodologies [38,39]. Presently, molecular classification techniques have been developed, typically involving the categorization of isolates based on the complete gene sequence of the nuclear small subunit 18S ribosomal RNA (rns). This approach facilitates the differentiation of Acanthamoeba spp. into 22 genotypes (T1–T22) and encompasses all known Acanthamoeba isolates discovered to date [40]. Most keratitis isolated belongs to the T4 group, followed by the T3 group in terms of frequency [35].

## 5. Pathogenesis

The principal manifestation of Acanthamoeba infection is keratitis; however, in immunocompromised individuals, it may lead to rare conditions such as granulomatous amoebic encephalitis (GAE), disseminated cutaneous amoebiasis, and visceral forms [5].

The pathogenesis of Acanthamoeba keratitis (AK) initiates with the trophozoite adhering to the corneal epithelium. This initial attachment is facilitated by a mannose-binding protein, which interacts with mannosylated glycoproteins present on corneal epithelial cells. This interaction triggers the release of MIP-133, a protease that induces the apoptosis and cytolysis of corneal epithelial cells, thus enabling an invasion from the cornea epithelium into the underlying stroma. Moreover, pathogenic isolates of Acanthamoeba are characterized by the production of further proteases which focus on the stromal layer of the cornea, including the plasminogen activator. Thus MIP-133 and other proteases produced by *Acanthamoeba* spp. May represent potential therapeutic targets for the management of AK [5,31].

It is plausible to hypothesize that contact lenses, being the primary risk factor for Acanthamoeba keratitis, may promote infection by facilitating the adherence of the microorganism to the corneal epithelium. In fact, the utilization of contact lenses leads to upregulation in the expression of mannosylated proteins on the corneal epithelium, thereby rendering the ocular surface more susceptible to trophozoite binding [5,31].

Studies have shown that corneal simple abrasions can induce changes in the expression of mannosylated proteins, potentially influencing susceptibility to Acanthamoeba keratitis (AK). Besides abrasions, numerous other factors contribute to the likelihood of Acanthamoeba infection. Badenoch et al., in their investigation, observed a heightened prevalence of the Gram-positive bacterium Corynebacterium xerosis in cultures obtained from the eyes of patients diagnosed with AK. They proposed that this bacterium could serve as both a nutritional source and a co-factor in the development of AK [41].

Moreover, Corynebacterium xerosis sets itself apart from other bacteria due to its notably elevated concentration of mannose within its cell wall, which could be crucial in promoting infection in the initial stages [42]. Similar to the distinctive microbiome observed in mammalian hosts, Acanthamoeba trophozoites harbor their own microbiome consisting of endosymbiotic bacteria. These endosymbionts have the potential to impact the pathogenicity and resistance of Acanthamoeba to therapeutic interventions. In a clinical investigation involving 23 patients diagnosed with Acanthamoeba keratitis (AK), it was revealed that more than half of the Acanthamoeba isolates retrieved from corneal lesions contained at least one endosymbiont [43]. Bacterial endosymbionts have demonstrated the ability to increase the in vitro cytopathogenicity of Acanthamoeba trophozoites [43,44].

The immune system serves a crucial function in protecting the organism from Acanthamoeba infection and in preventing its spread within the eye. In fact, despite numerous documented cases of Acanthamoeba keratitis (AK) in the literature, only six cases have progressed to involve infection in the posterior segments of the eye [45,46,47].

However, even when localized to the cornea, the infection can lead to significant damage, greatly impacting corneal function and vision. Both macrophages and neutrophils exhibit significant activity against Acanthamoeba, both in its trophozoite and cystic forms. In animal infection models, a severe exacerbation of the infection has been observed in the case of macrophage or neutrophil depletion [48,49,50]. The complement system, a pivotal component in resisting various microbial infections, also contributes to Acanthamoeba resistance. Trophozoites from non-pathogenic strains of Acanthamoeba are susceptible to cytolysis induced by complement activation through the alternative pathway [51]. However, pathogenic strains of Acanthamoeba evade complement-mediated lysis by expressing complement-regulatory proteins that deactivate the complement cascade [52]. Furthermore, there is significant evidence indicating that *Acanthamoeba* spp. can trigger responses from the adaptive immune system. Serological surveys reveal that around 90% of adults with no prior Acanthamoeba infections express serum IgG antibodies specific to Acanthamoeba antigens, and 50% of asymptomatic individuals exhibit T-cell responses to Acanthamoeba antigens [53,54,55]. Despite the presence of adaptive immunity, both pigs and Chinese hamsters remain susceptible to Acanthamoeba keratitis (AK), with no evidence suggesting that immunization prevents or alleviates corneal infection. However, the presence of secretory IgA antibodies provides a strong level of protection against the initial stages of corneal infection resulting from contact-lens contamination through the parasite [31].

The presence of dormant cysts in ocular tissues presents a potential risk for the recurrence of the infection [31]. In vitro experiments involving dexamethasone treatment have demonstrated a four- to sixfold increase in excystment and a doubling in the proliferation of emerging trophozoites [56]. Furthermore, dexamethasone activates Acanthamoeba trophozoites, significantly enhancing their cytolysis of corneal cells. These exacerbating effects have been observed in vivo, as evidenced through Chinese hamsters treated with dexamethasone exhibiting markedly more severe cases of Acanthamoeba keratitis [56]. These findings from in vitro and animal studies are supported by clinical research, which suggests that patients with Acanthamoeba keratitis (AK) who receive corticosteroid treatment before starting antimicrobial therapy tend to have significantly poorer outcomes compared to cases where steroid treatment is initiated after the commencement of antimicrobial therapy [57].

## 6. Clinical Symptoms

Establishing a diagnosis of AK through clinical presentation can be challenging due to the initial manifestations mimicking those of other corneal infections, and a high index of clinical suspicion is crucial. All age groups are susceptible to the condition, and there is no observed gender preference; however, a higher proportion of contact-lens wearers are female. AK has a chronic and indolent course, and the most frequent clinical manifestations documented in the scientific literature tend to be skewed towards the later stages of the disease, which aligns with the typical patient population seen in tertiary-care settings. Hence, a robust suspicion for Acanthamoeba infection should encompass patients exhibiting the aforementioned risk factors. Additional frequently noted attributes, such as a prolonged history, multiple physician consultations, negative cultures from scrapings, resistance to various antimicrobial agents (bacterial, fungal, and viral), and prior corticosteroid administration, are associated with the slow-progressing nature of the illness and the likelihood of a misdiagnosis or delayed diagnosis, rather than constituting inherent characteristics of the disease.

During the early stages of the disease, 75% to 90% of patients encounter diagnostic inaccuracies. Acanthamoeba patients frequently receive misdiagnoses, being erroneously labeled as herpetic keratitis, mycotic infection, or bacterial disease [58]. It has been described to mimic adenovirus conjunctivitis, with conjunctival follicular reactions and subepithelial corneal opacities [59,60], or herpes simplex keratitis, with epithelial defects and dendritic changes in the epithelium [60].

The study conducted by Daas et al. demonstrated that the accurate diagnosis in Germany was typically delayed in time by a range of 2.8 ± 4.0 months following the onset of initial clinical symptoms [58].

The early symptoms of AK are relatively nonspecific. Patients may only exhibit mild ocular discomfort, tearing, redness, or visual blurring. However, a distinguishing feature of AK, especially as the infection advances and stromal inflammation intensifies, is intense ocular pain [25,61]. Typically, the disease is unilateral, but in up to 7.5% of cases, the presentation may be bilateral [62,63]. Wilhelmus and coworkers investigated the prevalence and the characteristics of the binocular involvement of *Acantamoeba* keratitis over ten years at a single institution, demonstrating that the infection affects the eyes bilaterally in about 11% of cases, either with concurrent involvement of the cornea or in succession; the main feature they demonstrated is that it seems to be a typical complication of contact-lens wear [22].

In approximately 23% of cases [62,64,65,66], a mixed infection involving viruses, bacteria, or fungi is observed. Chuang et al. showed two cases of co-infection of Acanthamoeba with Pseudomonas (a contact-lens wearer presenting with a paracentral corneal ulcer and perineuritis) and microsporidia (presentation with multiple raised corneal lesions associated with epitheliitis): perineuritis in contact-lens wearers and epitheliitis in patients without risk factors are unusual presentations for AK and should raise suspicion of co-infection with other pathogens [67].

Tu et al. [68] defined five stages of AK disease with clinical findings based on the localization and the severity of the infection in the corneal layers: epitheliitis, epitheliitis with radial neuritis (the cause of the severe disproportionate pain), anterior stromal disease, deep stromal keratitis, or ring infiltrate (Figure 1).

The initial stages within the first two weeks of infection exhibit alterations in the epithelial and subepithelial layers, characterized by the term “chameleon-like epithelial changes.” These changes encompass features such as “dirty epithelium”, pseudodendritiform epitheliopathy, epithelial microerosions, and microcysts [64,65,66,70,71].

Limbitis is frequently observed from the early stages of Acanthamoeba keratitis (AK). This inflammatory response signifies the involvement of the limbus in the pathogenesis of the disease, possibly due to direct invasion by Acanthamoeba organisms or secondary to the host immune response.

As the condition progresses, characteristic infiltrations may appear along the corneal nerves, known as radial keratoneuritis or perineural stromal infiltrates, with a paracentral ring infiltrate, also known as the “Wessely immune ring”: the co-presence of these signs serves as a distinctive indicator of AK. The stromal infiltrates observed in acanthamoeba keratitis often present as multifocal, dot-like lesions, in contrast to the monofocal infiltrates typically observed in bacterial keratitis. While the characteristic stromal ring is evident in approximately 50% of AK cases, it can also be identified in bacterial corneal ulcers and fungal keratitis [61,68,72,73].

In AK, a variety of secondary signs and complications may develop beyond the primary symptoms of corneal pain, redness, and blurred vision. Secondary glaucoma can develop due to increased intraocular pressure, which can be caused by a variety of mechanisms: the edema of the trabecular meshwork, the endothelial cell dysfunction of the trabecular meshwork, fibrin and inflammatory cells blocking outflow through the trabecular meshwork or the Schlemm canal, and Peripheral anterior synechiae or posterior synechiae blocking outflow. Sterile anterior uveitis is another daunting complication which may result in broad-based anterior synechiae, iris atrophy, and lens changes that can lead to mature cataracts. The management of these cataracts, being complex, should be personalized according to the patient’s characteristics, and it is advisable to perform phacoemulsification as an elective procedure once the infection has been eradicated. Other potential complications include scleritis, chorioretinitis, and retinal vasculitis. In the majority of instances, these complications stem from inflammation rather than infection, as there are typically no discernible organisms present in the inflamed areas [71,74,75]. These secondary signs underscore the importance of timely diagnosis and appropriate management to mitigate long-term ocular damage and preserve vision.

## 7. Diagnosis

The diagnosis of Acanthamoeba keratitis is very challenging, and a timely and accurate identification is critical for initiating the appropriate treatment and preventing potential vision-threatening complications. Typically, the gold standard for corneal infections is the culture and isolation of the responsible pathogen through microbiological investigations. However, the sensitivity of these investigations for Acanthamoeba keratitis is quite low (cultural positivity is between 0% and 53%), so the diagnosis must rely on the combination of multiple factors: clinical suspicion, microbiological investigations, and corneal imaging [13,76]. Clinical suspicion should rise in the presence of symptoms and signs of AK in individuals with typical risk factors [61,62,77,78].

The following are the primary methodologies and instruments employed for diagnosing acanthamoeba keratitis (Table 1).

### 7.1. In Vivo Confocal Microscopy (IVCM)

IVCM is a helpful non-invasive diagnostic method that can be used during all the stages of the disease: it can be beneficial in the early investigation of the keratitis as well as in cases of deep infiltrates inaccessible to corneal scrapings and during ongoing anti-parasitic treatment (when trophozoites and cysts may reside in the deeper layers of the cornea) [79], and it can be compared to a noninvasive biopsy.

In clinics equipped with IVCM, it is considered the primary approach for the diagnosis of AK. IVCM holds critical statistical significance in identifying AK, demonstrating exceptional specificity (100%) and a notably high sensitivity, ranging from 85.3% to 100% [79,80,81].

This method offers immediate results, eliminating the need for the lengthy wait times associated with other diagnostic approaches.

In IVCM, *Acanthamoeba* spp. features depend on the stage of the disease. Cysts (dormant form) are easier to detect compared to the trophozoite form: they manifest as highly reflective, round structures in the deep stromal layers usually defined by their dual-layered walls, with diameters between 12 to 25 microns [61,82]. Occasionally, they have been described as contributing to a characteristic “starry sky” pattern [83,84,85].

Trophozoites (active form) appear as oval, serpentine, or pear-shaped formations, similar to leukocytes and keratocyte nuclei, and can pose challenges because of their resemblances in morphology [81,86,87]. They may present with bright spots and signet rings and a perineural patchy infiltrate with surrounding spindle-shaped materials [88,89,90].

Another alteration observable using IVCM is the loss of normal keratocyte morphology in the anterior stroma [91]: after damage, cellular transparency is reduced and the visibility of cellular structures increases [92].

Furthermore, after the use of topical steroids, cysts tend to form clusters [91,93], and this may refer to the mechanism of “biofilm formation” with poor prognosis [94].

There are several limitations associated with the use of in vivo confocal microscopy (IVCM) in diagnosing Acanthamoeba keratitis (AK). Firstly, access to IVCM is not widespread, limiting its availability for clinicians. Furthermore, proficiency in operating IVCM is crucial as it is operator-dependent and requires a significant learning curve to achieve expertise. Despite its potential, IVCM’s capability to image only a restricted area of the cornea per scan poses challenges, potentially leading to scans being taken in unaffected areas, thus compromising diagnostic accuracy. Additionally, the difficulty in detecting trophozoite forms and the potential masking of Acanthamoeba cysts by stromal corneal inflammation contribute to the risk of both false negatives and false positives. These limitations underscore the need for caution and complementary diagnostic approaches when utilizing IVCM for diagnosing AK, as this could lead to a late diagnosis.

### 7.2. Corneal Scraping

To confirm AK, it is essential to sample and analyze the appropriate material. A reliable diagnosis is only possible if amoebae are identified in corneal scrapings or biopsies. Acanthamoebae typically infiltrate deeply in the cornea and are not commonly found on its surface. Therefore, superficial swab or tear samples frequently yield negative results, especially in advanced disease stages or when patients have previously received antibiotic treatment.

It is important to obtain corneal samples prior to the initiation of any antimicrobial therapy [95], particularly if the drop-preservative benzalkonium chloride is present, and also the use of topical anesthetics such as proxymetacaine 0.5%, oxybuprocaine 0.4%, or povidone iodine have been shown to have toxic activity on Acanthamoeba cysts and trophozoites [96,97].

While culture tests boast high specificity (100%), their sensitivity is generally weak, varying from 7% to 70% in the different studies, depending on the sampling and the culture techniques employed [98].

Their effectiveness can also be compromised in cases of co-infection with another microbes, which may lead to an erroneous diagnosis of the keratitis [99]. Co-infection with a viral, fungal, or bacterial pathogen are present in approximately 23% to 55% of AK cases [100]. The most frequently encountered pathogens in co-infections include alpha-hemolytic Streptococcus spp., coagulase-negative *Staphylococcus* spp., *Bacillus* spp., *Corynebacterium* spp., *Staphylococcus aureus* spp., and *Streptococcus viridans* spp. [100].

The preferred specimen for diagnosing AK is a corneal scraping or biopsy preserved in 200 microliters of sterile saline solution. It has been reported that corneal sampling with cotton swabs [101], and even better with bezel needles [102], can more easily produce positive cultures than the use of a blade.

The plate-culture method remains the benchmark for detecting Acanthamoeba [103]. In this approach, the material (such as corneal scrapings, biopsies, a transport medium, contact lenses, swabs, etc.) is placed onto a non-nutrient agar plate. Bacteria serve as a key energy source for Acanthamoeba trophozoites: to support their proliferation, the agar plate is supplemented with a layer of heat-killed bacteria, typically Escherichia coli or Enterobacter cloacae. Daily monitoring of the plate is conducted to detect the presence of amoebae. To confirm a negative result reliably, the samples should be observed for a duration of up to three weeks.

Fresh specimens can also undergo staining to facilitate cyst detection; typically, these include calcofluor white (CFW); potassium hydroxide (KOH) wet mount; iodine, hematoxylin and eosin (H and E); Gomori methanamine silver (GMS); periodic acid-Shiff (PAS) stains; Gimenez stains; and lactofenol cotton blue [104,105].

Especially, CFW is a quick and easy staining technique to find amoebic cysts with great specificity (96%) but not very good sensitivity (71%) [106]; in contrast, KOH wet mount has very good sensibility (91.4%) and specificity (100%) [107].

In severe infections, amoebae density tends to be notably high, allowing for their detection via direct microscopy. These amoebae are distinguished by their prominent central nucleolus and hyaline pseudopodia, which feature characteristic hyaline protrusions known as acanthopodia. Trophozoites typically measure between 15 to 45 μm and exhibit an oval to elongated shape, showing a slow movement. Cysts are smaller in size (12 to 25 μm) and display a polygonal or star-shaped morphology with a double-layer wall [61].

### 7.3. PCR

Polymerase chain reaction (PCR) is a molecular technique utilized for the rapid amplification of genetic material (DNA) from Acanthamoeba in samples obtained from the ocular surface [108].

PCR analysis of sample material demonstrates the highest sensitivity, ranging from 84% to 100% according to previous studies [80,109,110,111], usually amplifying a fragment of the 18S rRNA gene [112,113,114] and it is also very fast, showing results already within a few hours [115].

PCR may exhibit a limitation whereby positive results can be obtained even in cases where non-viable Acanthamoeba genomes are present.

Molecular investigations enable the genotyping of the amoeba, revealing that the majority of Acanthamoeba keratitis (AK) cases worldwide are attributed to genotype T4, followed by the genotypes T3 and T11 [112].

### 7.4. Anterior Segment Optical Coherence Tomography (AS-OCT)

Recent studies have highlighted the effectiveness of AS-OCT in supporting AK diagnosis. While it cannot detect directly the Acanthamoeba cyst or trophozoites, AS-OCT can validate the clinical appearance of radial keratoneuritis as highly reflective bands oblique in the corneal stroma [116,117].

AS-OCT can also aid in differential diagnosis: according to Park et al., radial reflective bands in herpetic keratitis typically manifest as sub-epithelial, whereas in AK cases, they are located within the stromal layers (anterior to mid-stroma) [118]. As a non-invasive diagnostic tool with high-resolution capabilities, AS-OCT enables the visualization of radial keratoneuritis, particularly in cases where clinical observation is hindered by corneal edema and stromal infiltrates. The resolution of clinical perineuritis aligns closely with the disappearance of radial lines observed on AS-OCT, offering a valuable alternative for follow-up evaluations when confocal microscopy is unavailable [84,119].

### 7.5. Impression Cytology

Another less commonly employed diagnostic technique is impression cytology, which involves the retrieval of superficial corneal epithelial cells using nitrocellulose filters. Following this, the sample is stained with PAS/PAP or with CFW, allowing for the detection of Acanthamoeba cysts. Despite being relatively non-invasive and possessing high specificity for AK diagnosis, this approach requires the use of specialized stains and expertise in cytopathology, and it is not able to detect the presence of the cysts in the deep corneal layers [111,120,121].

## 8. Treatment

The therapeutic regimen for AK is frequently prolonged and arduous. Although the trophozoite stage of Acanthamoeba spp. is responsive to various treatments, the cystic form exhibits significant resistance to drugs and can endure for months [85].

Typically, treatment involves the utilization of a combination of different drug active substances (Table 2).

### 8.1. Conservative Treatment

Administering a topical biguanide, like polyhexamethylene biguanide (PHMB) 0.02–0.08% [122], or chlorhexidine 0.02–0.06% [123], along with a topical diamidine, such as propamidine isethionate 0.1% [62,83,84,85,124,125,126,127,128], constitutes the principal initial treatment.

Typically, therapy involves the application of topical eye drops, initially administered every hour for the first few days. To improve the penetration of topical medications and physically eliminate trophozoites and cysts confined to the corneal epithelium, corneal debridement may be conducted.

Treatment usually spans a duration of 3–4 weeks at a minimum, with the frequency of eye-drop administration reducing to every three hours following the initial days. According to most authors, it is advisable to prolong treatment for several months, potentially up to a year, to thwart the occurrence of reinfection or disease recurrence [32].

The proposed treatment plan by Varacalli et al. for AK involves an aggressive initial approach with hourly topical eye drops containing PHMB 0.02% and propamidine isethionate 0.1%. Following this, the regimen is gradually tapered to maintenance therapy, administered 3–4 times per day for 6 weeks. A stable clinical examination after a 2-week period without antiamoebic medication helps mitigate medication toxicity and assesses for any persistence of trophozoites or cysts. Persistent infection necessitates repeating the treatment protocol (Figure 2) [23].

#### 8.1.1. Biguanides

Biguanides demonstrate efficacy as antimicrobial agents by exerting both cystocidal and cystostatic effects. Their positively charged molecules adhere to and permeate the amoebas, enhancing membrane permeability and resulting in the pathogen’s demise [32]. Polyhexamethlyene biguanide (PHMB) and chlorhexidine are two biguanide compounds consistently proven effective in drug treatment.

At a diluted concentration of 0.02%, PHMB, commonly employed as a disinfectant for pools, displays effective cysticidal activity against multiple variants of the pathogen [129]. While chlorhexidine exhibits slightly diminished cysticidal effectiveness relative to PHMB, its smaller molecular size facilitates deeper penetration into the corneal stroma, possibly rendering it an efficient substitute [130]. The primary adverse effects associated with biguanide treatment may include increased intraocular pressure and toxic keratopathy, potentially necessitating a reduction in drug dosage or a transition to an alternative therapeutic regimen [32].

A multicenter phase-3 randomized trial conducted by Dart et al. analyzed 127 patients suffering from AK keratitis and suggested that PHMB 0.08% monotherapy could be equally effective compared to the widely used dual therapy of PHMB 0.02% + propamidine in populations with similar disease severity [131].

Some studies have indicated that using 0.02% PHMB or 0.02% chlorhexidine alone for the initial treatment of AK in early disease is comparable in efficacy to combination therapies, which include both a biguanide and a diamidine; this treatment strategy has demonstrated favorable cure rates and is appealing because administering a single medication can enhance patient adherence and reduce costs compared to combined therapy [123,132].

#### 8.1.2. Aromatic Diamidines

Aromatic diamidines like propamidine and hexamidine are commonly employed alongside biguanides in the treatment of AK. One proposed mechanism of action for aromatic diamidines involves binding to the parasite’s DNA, thereby inhibiting its growth [130]. Although they possess cystostatic activity, they lack cystocidal properties and hence cannot be used as standalone therapy [121]. Propamidine, one of the earliest treatments discovered for AK, has been incorporated into numerous combination therapies. However, reports indicate instances of Acanthamoeba developing resistance to propamidine, underscoring the preference for its use alongside a biguanide [121,133,134].

Both biguanides and diamidines have the potential to induce corneal toxicity, often resulting in corneal epitheliopathy.

#### 8.1.3. Additional Medicaments

Neomycin indirectly affects Acanthamoeba by reducing bacterial food for trophozoites and preventing bacterial superinfection, thereby eliminating the trophozoite form. However, it lacks significant cysticidal activity and cannot be used in monotherapy due to the risk of neomycin-resistant temperature-sensitive mutants emerging and the inherent resistance of cysts. Nonetheless, when incorporated into a “triple therapy” with PHMB and propamidine, neomycin has effectively treated many AK patients [135,136].

In vitro studies suggest that low levels of benzalkonium chloride (BAK) and povidone iodine demonstrate notable anti-Acanthamoeba activity [97,137].

Diverse types of medications have shown potential in managing AK.

Systemic antifungal agents, including voriconazole and posaconazole, may offer efficacy against the cystic phase of Acanthamoeba spp. by impeding the synthesis of ergosterol, a crucial component of the Acanthamoeba spp. cell membrane, presenting a possible treatment choice to halt cyst formation [84,138]. Miltefosine, an oral drug employed in treating leishmaniasis and amoebic infections, has been utilized in AK therapy. Despite its current costliness or accessibility challenges, it has proven effective in managing AK [32,139].

Alternative treatments, like topical extract from the tea tree, could offer potential as complementary therapies: it displayed full efficacy against both trophozoite and cyst stages of Acanthamoeba in laboratory settings. While additional investigation is necessary, this semi-in vivo examination suggests that tea tree oil administered as eye drops may have the ability to eradicate amoebae both in superficial and deep corneal layers [140].

#### 8.1.4. Steroids

The utilization of corticosteroids to alleviate the intense inflammation associated with AK remains a topic of debate due to the drawback of concealing clinical indicators, which could potentially encourage cyst formation and escalate trophozoite proliferation [141].

During the early stages of AK, corticosteroids generally play no role. However, in patients with prolonged treatment and persistent severe corneal inflammation, limbitis or scleritis, reports suggest that topical corticosteroids may assist in disease resolution alongside systemic non-steroidal anti-inflammatory drugs (NSAIDs) [26,142]. If administered, corticosteroids should always be paired with simultaneous antiamoebic treatment.

#### 8.1.5. Crosslinking

Corneal collagen cross-linking (CXL) represents a relatively recent therapeutic avenue explored in AK management. Although laboratory investigations conducted by Kashiwabuchi et al. [143] and Del Buey et al. [144] have demonstrated the absence of any amoebicidal impact of riboflavin in conjunction with UVA exposure, clinical case reports paint a more encouraging perspective [145,146]. Nevertheless, there exists insufficient evidence at present to substantiate its application in the context of AK [147,148].

### 8.2. Surgical Treatment

Corneal transplants, including deep anterior lamellar keratoplasty (DALK) or full-thickness-penetrating keratoplasty (PK), may be necessary for AK cases resistant to medical therapy. However, the precise role and timing of keratoplasties in AK remain uncertain. Sarnicola et al. suggested that performing early DALK within 30–60 days of symptom onset, alongside antiamoebic treatment before, during, and after surgery, can effectively eliminate infection and significantly enhance final post-operative BCVA [149]. Despite DALK carrying lower risks of rejection and graft failure compared to PK, it is technically more challenging and may be less effective in clearing the infection, particularly in inflamed eyes or advanced disease stages [149]. Full-thickness PK is the foremost surgical option to prevent scleral extension [149,150]. It is also recommended for corneal perforation and severe corneal abscesses [32,62,142]. However, executing PK in severe AK cases demands careful assessment, given the typically inferior prognosis linked with transplants in these eyes [21]: to appropriately size the corneal transplant, surgeons must strike a balance between the risk of rejection and failure associated with larger grafts and the need to remove all affected tissue to minimize recurrence. Recurrence post-keratoplasties (PK or DALK) most commonly manifest within the initial two weeks post-operation, but late recurrences are also plausible [21]. Di Zazzo et al. showed that the recurrence rate of AK following different types of corneal keratoplasties varies: 16.8% for PK, 19% for DALK, and 9.5% for optical keratoplasties. This recurrence is attributed to the reactivation of dormant cysts in the recipient bed, leading to the colonization of the donor cornea [150]. To minimize the risk of disease recurrence, a recommended target is to achieve a 1 mm margin of healthy tissue [151]. Moreover, it is advised to maintain antiamoebic treatment for several months following the surgical procedure [32,62].

Optical keratoplasties (PK or DALK), performed as elective surgeries for corneal scarring after the AK resolution (Figure 3 and Figure 4), obtain better outcomes in terms of graft survival and BCVA compared to therapeutic *à chaud* keratoplasties [132].

A review of 359 eyes showed that 94% of optical keratoplasty patients had a clear graft at follow-up, compared to 73% for PK and 84% for DALK. Furthermore 40% achieved a visual acuity exceeding 20/30, while 21% and 25% of PK and DALK recipients, respectively, reached this level. Additionally, optical keratoplasties had the lowest recurrence rate of infection among the procedures [150]. Therefore, it is advisable to prioritize medical treatments whenever possible and defer surgical interventions until Acanthamoeba spp. eradication is achieved.

Furthermore, in conjunction with DALK and PK, amniotic membrane transplantation presents an alternative avenue for facilitating full corneal restoration. It effectively addresses inflammation and fosters the healing of both stromal and epithelial tissues, particularly in instances of advancing stromal lesions and persistent epithelial defects. However, attaining complete recovery may necessitate multiple amniotic membrane transplantations, and it may still lead to the requirement a corneal transplant [32,85,125,150].

## Figures and Tables

**Figure 1 microorganisms-12-00758-f001:**
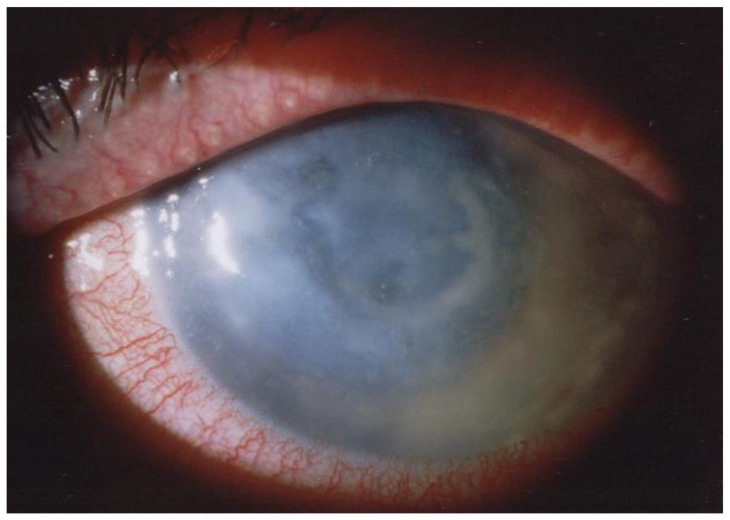
AK with deep stromal keratitis [69].

**Figure 2 microorganisms-12-00758-f002:**
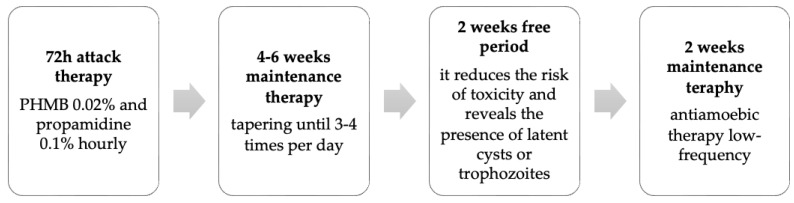
Therapeutic strategy for Acanthamoeba keratitis with PHMB and propamidine.

**Figure 3 microorganisms-12-00758-f003:**
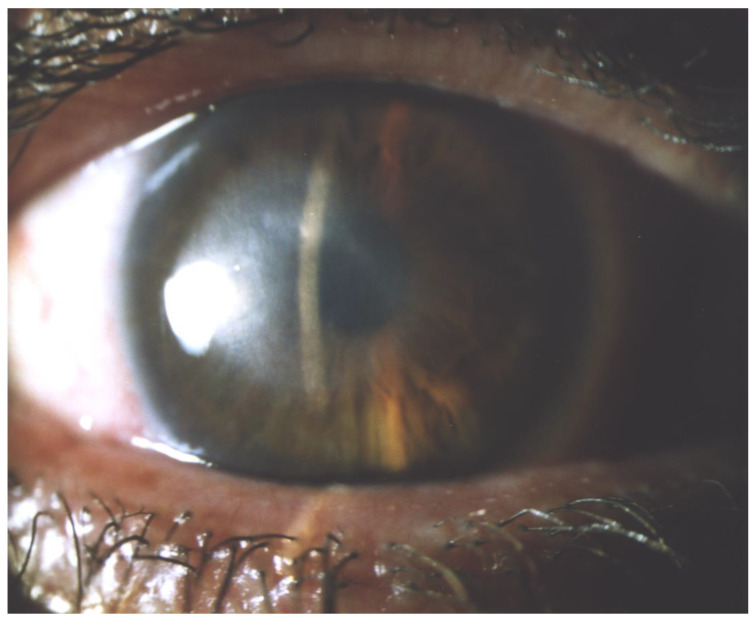
Corneal leucoma post AK, pre perforating keratoplasty.

**Figure 4 microorganisms-12-00758-f004:**
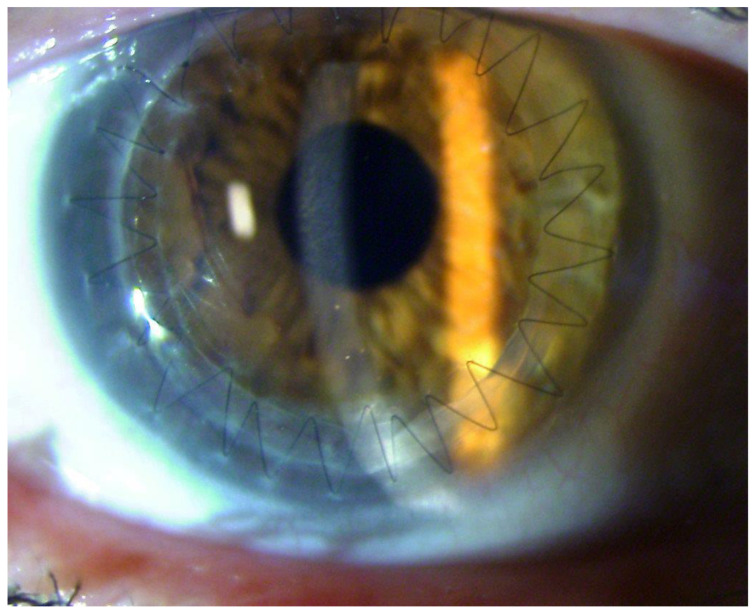
Post Perforating keratoplasty to treat corneal leucoma.

**Table 1 microorganisms-12-00758-t001:** Diagnosis of AK: advantages and disadvantages.

Diagnostic Tool	Advantages	Disadvantages
IVCM	Primary approach. High specificity and sensitivity, non-invasive and rapid. It can be used during all the stages of the disease.	Not widespread. It is operator-dependent and it requires a significant learning curve. It analyzes only a restricted area of the cornea per scan. It has difficulty in detecting trophozoite forms and it can mask Acanthamoeba cysts by stromal corneal inflammation.
COLTURE	High specificity and good sensitivity.	Acanthamoebae typically infiltrate deeply in the cornea and are not commonly found on its surface. Misdiagnosis in case of co-infection with another microbes.
PCR	Quick and very sensitive.	Positive results can be obtained even in cases where non-viable Acanthamoeba genomes are present.
AS-OCT	Non-invasive and useful for evaluating the radial keratoneuritis and the differential diagnosis.	It cannot detect directly the Acanthamoeba cyst or trophozoites.
IMPRESSION CITOLOGY	High specificity and relatively non-invasive.	It requires the use of specialized stains and expertise in cytopathology. It is not able to detect the presence of the cysts in the deep corneal layers.

**Table 2 microorganisms-12-00758-t002:** Treatment of AK.

Treatment	Type of Treatment	When to Use It
Medical treatment	Biguanides (PHMB, chlorhexidine)	At a diluted concentration of 0.02% is the first line of treatment (alone or in combination with diamidines)
	Aromatic diamidines (propamidine, hexamidine)	Alongside biguanides
	Neomycin	Only with PHMB and propamidine
	Antifungical agents (voriconazole, posaconazole)	Efficacy against only the cystic phase
	Miltefosine	Effective, but expensive and not easily accessible
	Extract from tea tree	Only in laboratory settings
	Steroids	No role
	NSAIDs	With severe corneal inflammation, limbitis, or scleritis
Parasurgical treatment	Cross-linking of corneal collagen	Non-universal consensus on amoebicidal impact of riboflavin + UVA
Surgical treatment	DALK	Cases resistant to medical therapy (uncertain role and timing)
	PK	

## Data Availability

No new data were created or analyzed in this study. Data sharing is not applicable to this article.

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
