# Peer review of "Understanding Acanthamoeba Keratitis: An In-Depth Review of a Sight-Threatening Eye Infection"

_microorganisms, 2024, doi:10.3390/microorganisms12040758_

Round 1

Reviewer 1 Report

Comments and Suggestions for Authors

Thank you for your trust and entrusting me with the role of reviewer. Due to the increasing use of contact lenses, the number of Acanthamoeba corneal infections is also increasing. The publication presents a comprehensive approach to all aspects of Acanthamoeba keratitis. It includes epidemiological characteristics, risk factors, pathophysiology, clinical symptoms, diagnostic tests and therapeutic procedures. It is a very good source of knowledge for both young doctors and specialists. The work is well constructed. References cited correctly.  

Author Response

We sincerely appreciate the reviewer's comments and his esteemed opinion.

Kind Regards

Reviewer 2 Report

Comments and Suggestions for Authors

The article "Understanding Acanthamoeba Keratitis: An In-Depth Review of a Sight-Threatening Eye Infection" presents a comprehensive analysis of Acanthamoeba keratitis (AK), a rare but potentially severe corneal infection caused by the Acanthamoeba parasite. 

The article lacks any tables and figures. Please create several tables and figures/pictures. Maybe pictures of patients from your own experience?

Please ensure that the genus and species names are written with italics, that is the general accepted convention.  

Authors should try to find newer studies regarding the incidence of AK for the epidemiology sections.

While the review extensively covers diagnosis and treatment, there's less emphasis on prevention strategies, which are crucial in mitigating the risk of AK.

Line 321: How do you explain the presence of anterior synechiae in AK?

Line 322: Which mechanism causes ocular hypertension: is it pre-trabecular, trabecular, or post-trabecular? How does one assess glaucoma shift in eyes with severe keratitis? 

Line 323/324: How do mature cataracts form in AK? How does one operate a mature cataract in an eye with acute AK or central large leukoma post-AK?

Line 324: Is sterile uveitis connected to anterior synechiae, secondary hypertension, and mature cataract formation? Is sterile uveitis an exogenous or an endogenous uveitis?

Line 420: please corect "idroxide" with hidroxide

Line 481: where is table 1

Line 577: In case of documented AK or highly suspect AK, what is the therapy sequence?

Line 605-606: Are there any studies regarding dormant cysts persistance and post-keratoplasty reccurence?

Line 611-613: Please comment why keratoplasties after AK resolution obtain better outcomes than keratoplasties in active infection?

Author Response

REV 2

We sincerely appreciate the reviewer's comments and his esteemed opinion.

R1: The article lacks any tables and figures. Please create several tables and figures/pictures. Maybe pictures of patients from your own experience?

A1: We thank the reviewer for the comment. We have included several tables and figures as requested.

R2: Please ensure that the genus and species names are written with italics, that is the general accepted convention.

A2: We thank the reviewer for the comment. We have modified as requested.

Among the various Acanthamoeba species implicated in keratitis, Acanthamoeba castellanii stands out as the most commonly associated, with Acanthamoeba polyphaga and Acanthamoeba hatchetti also frequently identified. Other species, including Acanthamoeba culbertsoni, Acanthamoeba rhysodes, Acanthamoeba lugdunesis, Acanthamoeba quina, and Acanthamoeba griffini, have also been described in relation to keratitis cases.

R3: Authors should try to find newer studies regarding the incidence of AK for the epidemiology sections.

A3: We thank the reviewer for the comment. We have included newer studies regarding the incidence of AK.

Randag et al have conducted a comprehensive assessment of Acanthamoeba keratitis (AK) incidence within the Netherlands spanning from 2009 to 2015. Their study revealed a notable escalation in AK cases, rising from 16 instances in 2009 to 45 occurrences in 2015. This increase signifies a substantial rise in AK incidence, translating to approximately 1 case per 21,000 contact lens wearers in 2015. [DOI: 10.1371/journal.pone.0222092]

Nielsen et al conducted a retrospective study at the tertiary ophthalmology department of Aarhus University Hospital in Denmark. From 1994 to 2018, they observed a notable increase in the incidence of Acanthamoeba keratitis (AK), with rates escalating from 0.13 cases per million per year during the initial five years to 2.7 cases per million per year in the last five years of the study period. [https://doi.org/10.1111/aos.14337]

R4: While the review extensively covers diagnosis and treatment, there's less emphasis on prevention strategies, which are crucial in mitigating the risk of AK.

A4: We thank the reviewer for the comment. We have modified the text as requested.

Understanding the primary risk factors associated with Acanthamoeba keratitis (AK) is pivotal for infection prevention. Mitigating risky behaviors, enhancing water sanitation practices, and promptly seeking care at specialized corneal disease centers can significantly contribute to prevent AK infection or its most severe complications. [DOI: 10.1371/journal.pone.0222092]

R5: Line 321: How do you explain the presence of anterior synechiae in AK?

A5: We thank the reviewer for the comment. We have clarified the information in the text as requested.

In AK, a variety of secondary signs and complications may develop beyond the primary symptoms of corneal pain, redness, and blurred vision. Secondary glaucoma can develop due to increased intraocular pressure which can be caused by a variety of mechanisms: edema of the trabecular meshwork, endothelial cell dysfunction of the trabecular meshwork, fibrin and inflammatory cells blocking outflow through the trabecular meshwork or Schlemm canal and Peripheral anterior synechiae or posterior synechiae blocking outflow.Sterile anterior uveitis is another daunting complication which may result in broad-based anterior synechiae, iris atrophy and lens changes that can lead to mature cataract. The management of these cataracts, being complex, should be personalized according to the patient's characteristics and it is advisable to perform phacoemulsification as an elective procedure one the infection has been eradicated. Other potential complications include: scleritis, chorioretinitis, and retinal vasculitis. In the majority of instances, these complications stem from inflammation rather than infection, as there are typically no discernible organisms present in the inflamed areas[72,75,76]. These secondary signs underscore the importance of timely diagnosis and appropriate management to mitigate long-term ocular damage and preserve vision.

R6: Line 322: Which mechanism causes ocular hypertension: is it pre-trabecular, trabecular, or post-trabecular? How does one assess glaucoma shift in eyes with severe keratitis?

A6: We thank the reviewer for the comment. We have clarified the information in the text as requested.

In AK, a variety of secondary signs and complications may develop beyond the primary symptoms of corneal pain, redness, and blurred vision. Secondary glaucoma can develop due to increased intraocular pressure which can be caused by a variety of mechanisms: edema of the trabecular meshwork, endothelial cell dysfunction of the trabecular meshwork, fibrin and inflammatory cells blocking outflow through the trabecular meshwork or Schlemm canal and Peripheral anterior synechiae or posterior synechiae blocking outflow.Sterile anterior uveitis is another daunting complication which may result in broad-based anterior synechiae, iris atrophy and lens changes that can lead to mature cataract. The management of these cataracts, being complex, should be personalized according to the patient's characteristics and it is advisable to perform phacoemulsification as an elective procedure one the infection has been eradicated. Other potential complications include: scleritis, chorioretinitis, and retinal vasculitis. In the majority of instances, these complications stem from inflammation rather than infection, as there are typically no discernible organisms present in the inflamed areas[72,75,76]. These secondary signs underscore the importance of timely diagnosis and appropriate management to mitigate long-term ocular damage and preserve vision.

R7: Line 323/324: How do mature cataracts form in AK? How does one operate a mature cataract in an eye with acute AK or central large leukoma post-AK?

A7: We thank the reviewer for the comment. We have clarified the information in the text as requested.

In AK, a variety of secondary signs and complications may develop beyond the primary symptoms of corneal pain, redness, and blurred vision. Secondary glaucoma can develop due to increased intraocular pressure which can be caused by a variety of mechanisms: edema of the trabecular meshwork, endothelial cell dysfunction of the trabecular meshwork, fibrin and inflammatory cells blocking outflow through the trabecular meshwork or Schlemm canal and Peripheral anterior synechiae or posterior synechiae blocking outflow.Sterile anterior uveitis is another daunting complication which may result in broad-based anterior synechiae, iris atrophy and lens changes that can lead to mature cataract. The management of these cataracts, being complex, should be personalized according to the patient's characteristics and it is advisable to perform phacoemulsification as an elective procedure one the infection has been eradicated. Other potential complications include: scleritis, chorioretinitis, and retinal vasculitis. In the majority of instances, these complications stem from inflammation rather than infection, as there are typically no discernible organisms present in the inflamed areas[72,75,76]. These secondary signs underscore the importance of timely diagnosis and appropriate management to mitigate long-term ocular damage and preserve vision.

R8: Line 324: Is sterile uveitis connected to anterior synechiae, secondary hypertension, and mature cataract formation? Is sterile uveitis an exogenous or an endogenous uveitis?

A8: We thank the reviewer for the comment. We have clarified the information in the text as requested.

In AK, a variety of secondary signs and complications may develop beyond the primary symptoms of corneal pain, redness, and blurred vision. Secondary glaucoma can develop due to increased intraocular pressure which can be caused by a variety of mechanisms: edema of the trabecular meshwork, endothelial cell dysfunction of the trabecular meshwork, fibrin and inflammatory cells blocking outflow through the trabecular meshwork or Schlemm canal and Peripheral anterior synechiae or posterior synechiae blocking outflow.Sterile anterior uveitis is another daunting complication which may result in broad-based anterior synechiae, iris atrophy and lens changes that can lead to mature cataract. The management of these cataracts, being complex, should be personalized according to the patient's characteristics and it is advisable to perform phacoemulsification as an elective procedure one the infection has been eradicated. Other potential complications include: scleritis, chorioretinitis, and retinal vasculitis. In the majority of instances, these complications stem from inflammation rather than infection, as there are typically no discernible organisms present in the inflamed areas[72,75,76]. These secondary signs underscore the importance of timely diagnosis and appropriate management to mitigate long-term ocular damage and preserve vision.

R9: Line 420: please corect "idroxide" with hidroxide

A9: We thank the reviewer for the comment. We have modified the text as requested.

R10: Line 481: where is table 1

A10: We thank the reviewer for the comment. We have included the table 1 in the text.

R11: Line 577: In case of documented AK or highly suspect AK, what is the therapy sequence?

A11: We thank the reviewer for the comment. We Have included a figure to explain it better.

R12: Line 605-606: Are there any studies regarding dormant cysts persistance and post-keratoplasty reccurence?

A12: We thank the reviewer for the comment. We modified the text as requested.

Di Zazzo et al showed that the recurrence rate of AK following different types of corneal keratoplasties varies: 16.8% for PK, 19% for DALK and 9.5% for optical keratoplasties. This recurrence is attributed to the reactivation of dormant cysts in the recipient bed, leading to colonization of the donor cornea.

R13: Line 611-613: Please comment why keratoplasties after AK resolution obtain better outcomes than keratoplasties in active infection?

A13: We thank the reviewer for the comment. We modified the text as requested

A review of 359 eyes showed that 94% of optical keratoplasty patients had a clear graft at follow-up, compared to 73% for PK and 84% for DALK. Furthermore 40% achieved a visual acuity exceeding 20/30, while 21% and 25% of PK and DALK recipients respectively reached this level. Additionally, optical keratoplasties had the lowest recurrence rate of infection among the procedures